# The Pivotal Role of Presepsin in Assessing Sepsis-Induced Cholestasis

**DOI:** 10.3390/diagnostics14161706

**Published:** 2024-08-06

**Authors:** Maria Iuliana Ghenu, Dorin Dragoș, Maria Mirabela Manea, Andra-Elena Balcangiu-Stroescu, Dorin Ionescu, Lucian Negreanu, Adelina Vlad

**Affiliations:** 11st Department Medical Semiology, Faculty of Medicine, Carol Davila University of Medicine and Pharmacy, 020021 Bucharest, Romania; maria.ghenu@drd.umfcd.ro (M.I.G.); dorin.ionescu@umfcd.ro (D.I.); 21st Internal Medicine Clinic, Emergency University Hospital, 050098 Bucharest, Romania; 36th Department Clinical Neurosciences, Faculty of Medicine, Carol Davila University of Medicine and Pharmacy, 020021 Bucharest, Romania; maria.manea@umfcd.ro; 4Neurology Clinic, National Institute of Neurology and Neurovascular Diseases, 041915 Bucharest, Romania; 5Faculty of Dental Medicine, Carol Davila University of Medicine and Pharmacy, 010221 Bucharest, Romania; andra.balcangiu@umfcd.ro; 6Nephrology Clinic, Emergency University Hospital, 050098 Bucharest, Romania; 75th Department Internal Medicine, Faculty of Medicine, Carol Davila University of Medicine and Pharmacy, 020021 Bucharest, Romania; lucian.negreanu@umfcd.ro; 8Gastroenterology Clinic, Emergency University Hospital, 050098 Bucharest, Romania; 9Department of Functional Sciences I/Physiology 2, Faculty of Medicine, Carol Davila University of Medicine and Pharmacy, 050474 Bucharest, Romania; adelina.vlad@umfcd.ro

**Keywords:** presepsin, sepsis, sepsis-induced cholestasis, cholestasis-related parameters, inflammation, CD14

## Abstract

Background: The serum levels of presepsin correlate with parameters indicating cholestasis in sepsis; however, the probability and significance of this association remain uncertain. We aimed to ascertain whether infection, as signaled by presepsin levels, is the primary determinant of elevated biliary parameters in sepsis. Methods: A unicenter, retrospective study included 396 COVID-free emergency-admitted patients, in which presepsin level was determined. Presepsin, neutrophil count, leukocyte count, C reactive protein, and fibrinogen evaluated the septic/inflammatory state. The statistically significant factors associated with cholestasis, ALT, and AST were analyzed by Fisher’s exact test and Spearman regression with Bonferroni’s correction. Results: Presepsin emerged as the most likely variable correlated with all cholestasis markers: alkaline phosphatase (*p* = 7 × 10^−8^), gamma-glutamyl transferase (*p* = 5 × 10^−10^), and conjugated bilirubin (*p* = 4 × 10^−15^). Thrombocyte count, C reactive protein, age, creatinine, urea, lactate, and blood pressure, were associated with only one or two of these markers. Conclusions: In a sepsis setting, the increase in cholestasis-related parameters is associated with presepsin with a higher probability than hemodynamic, inflammatory, or coagulation-related variables. Determining this robust link between sepsis and cholestasis could eliminate unnecessary imaging procedures in critically ill patients, enabling clinicians to focus efforts on addressing the primary infectious cause.

## 1. Introduction

Sepsis is a maladaptive host reaction to infection resulting in biochemical, physiological, and biological alterations that might lead, via an uncontrolled inflammatory response, to multiple organ dysfunctions with life-threatening potential [1,2]. Cluster of differentiation 14 (CD14) is a glycoprotein known for binding lipopolysaccharides (LPSs) in Gram-negative bacterial cells. However, it can also act as a receptor for peptidoglycan from Gram-positive bacteria or other microbial products [3]. CD14 is primarily expressed on monocytes and macrophages, and to a lesser extent on neutrophils [4,5]. It exists in two forms: membrane-bound (mCD14) and soluble (sCD14) [6,7]. Presepsin, a subtype of sCD14 associated with sepsis severity [8], is valuable for diagnosing bacterial infections [6,7].

Presepsin is a biomarker useful for the early diagnosis of sepsis [9,10,11,12,13], with a predictive ability for septic shock and organ dysfunction severity independent of procalcitonin [14] and an efficiency comparable to that of procalcitonin [9,15,16,17]. Presepsin has the advantage of increasing earlier than procalcitonin, allowing for rapid detection, and is therefore particularly suited for use in the emergency department and critical care setting [18]. Some studies suggest that, compared to procalcitonin, presepsin has a higher sensitivity [6], a higher positive predictive value [19], and a similar area under the curve [20] for sepsis diagnosis, with a better predictive ability, particularly after hepato-biliary-pancreatic surgery [21]. Presepsin is also able to predict sepsis-related mortality in various settings [22,23], including intensive care units [24] and acute kidney injury [25]. It is a better predictor of sepsis-related mortality than procalcitonin [26], with similar sensitivity but better specificity [27].

Gamma-glutamyl transferase (GGT) and alkaline phosphatase (AlkPh) are generally considered markers of cholestasis; however, isolated elevations of either of them can occur in liver-unrelated conditions [28]. Cholestatic liver diseases are characterized by GGT and AlkPh elevations, typically but not always accompanied by rises in conjugated bilirubin (ConjBil), and to a lesser extent if at all, in alanine aminotransferase (ALT) and aspartate aminotransferase (AST) [28]. The onset of cholestasis is linked to increased mortality in individuals experiencing sepsis [29,30]. LPS-induced sepsis may generate cholestasis by diverse mechanisms. It may interfere with membrane pumps’ activity, which diminishes the uptake of bile acids and bilirubin into hepatocytes and their subsequent secretion into bile [31,32,33]. Additionally, LPS-induced sepsis may affect aquaporins, thereby decreasing canalicular membrane permeability for water, with impaired bile secretion [34]. It also hinders the function of nuclear receptors involved in inflammatory responses, such as the Farnesoid X receptor, hepatocyte nuclear factor 1, retinoid X receptor-retinoid acid receptor, and footprint B binding protein [31,32,35,36]. Another mechanism consists of activating the phosphatidylinositol-3-kinase signaling pathways, resulting in a decreased expression of multidrug resistance protein 2 (MRP2) [37] and bile salt export pump (BSEP) [32,38]. Toll-like receptor 4 is activated by LPS-induced sepsis as well, with subsequent increased cytokine production [7,39], a reduced expression of organic anions transporting polypeptides 4 (OATP4) [32], and a decreased level of type 3 inositol trisphosphate receptor, involved in bile formation [40]. Finally, the pro-inflammatory state of sepsis, with elevated IL-6 levels resulting in reduced sodium-taurocholate co-transporting polypeptide (NTCP) and MRP2 transcription, may lead to cholestasis [32].

As both presepsin and cholestasis-related parameters increase during sepsis, a positive correlation between presepsin level and biliary parameters (ConjBil, GGT, and AlkPh) in septic patients is to be expected. However, a similar correlation exists in non-septic patients, as presepsin clearance occurs not only via the kidney but also by the liver–biliary system [41]. Consequently, sepsis-unrelated cholestasis is associated with high presepsin levels [41], while parallel increases in both presepsin and biliary enzymes (AlkPh and GGT) were noticed in sepsis-free patients with normal renal function [42].

The present study aims to determine whether the alteration in cholestasis-related parameters commonly observed in septic states is primarily caused by the infection itself or by the inflammation it triggers. Our working hypothesis is that infection, as reflected by presepsin levels, is the main determinant of the increased biliary parameters in sepsis. Recognizing this strong sepsis-leading-to-cholestasis relation may help avoid unnecessary imaging procedures, particularly in critically ill patients, allowing clinicians to focus their efforts on treating the underlying infectious process.

## 2. Materials and Methods

### 2.1. Study Design and Ethical Issues

The research was conducted in the Department of Internal Medicine, University Emergency Hospital Bucharest, as a unicenter retrospective study, in respect of the Declaration of Helsinki. The study protocol was approved by the Ethics Committee of the University Emergency Hospital Bucharest, registered under the number 5843/1 February 2021. Written informed consent was obtained from all participants in the study, following General Data Protection Regulation (GDPR) requirements. All patients admitted to our hospital are asked to sign an informed consent form, allowing the results from their clinical examination, laboratory tests, and imaging investigations to be used for research purposes. If patients are unable to sign due to cognitive impairment, impaired consciousness, or motor function limitations, a close relative is asked to sign on their behalf.

### 2.2. Study Population

The study included all COVID-free patients, irrespective of age or diagnosis, admitted on an emergency basis to the 1st Internal Medicine and Nephrology departments of University Emergency Hospital Bucharest between 1 December 2017 and 31 December 2020, for whom presepsin level was determined at the emergency department within 24 h before admission. Exclusion criteria comprised: (1) extreme levels of leukocyte count, either too high (>70.000/μL) or too low (<1.000/μL); (2) the coexistence of another condition (besides sepsis) that might increase bilirubin in patients with high levels of bilirubin—this includes various disorders (cirrhosis, liver metastases, right-sided cardiac failure) or medication. A search on https://www.drugs.com/ (accessed on 22 April 2022) was performed for each medicine the patients were given during their hospital stay. Pharmacological agents found to induce cholestasis in at least 1% of treated patients were considered potential causes of ConjBil increase: rosuvastatin (in seven patients), atorvastatin (in four patients), azithromycin (in two patients), moxifloxacin (in four patients), cefepime (in one patient); (3) dilated bile ducts by ultrasound or computerized tomography; (4) a lack of information about the serum level of conjugated bilirubin.

A total of 544 patients were enrolled in the study. Applying the exclusion criteria led to the removal of 148 patients, with the entire process being summarized as a flow diagram in Figure 1.

We had a total of 396 patients included in the study, 214 women and 182 men, with ages between 24 and 98 years, mean ± standard deviation of 71.74 ± 13.47 years, a median of 73 years, and an interquartile interval of 64 to 82 years. The patients were divided into two groups according to ConjBil level: normal (*n* = 253) or increased (*n* = 143).

### 2.3. Variables Assessment/Definition

Sepsis was defined according to The Third International Consensus Definitions for Sepsis and Septic Shock [1,2]. Presepsin concentration was determined by chemiluminescent enzyme immunoassay using an automatic analyzer PATHFAST (manufactured by LSI Medience Corporation, Tokyo, Japan; authorized representative: Mitsubishi Chemical Europe GmbH, Duesseldorf, Germany) and expressed in pg/mL. The reference levels were: <200 = L (low), 300–500 = H (high), 500–1000 = 2H, >1000 = 3H. Values exceeding 20,000 were recorded as “>20,000” without specific numerical values provided; this applied to seven patients. To facilitate computations, a value of 20,001 was assigned to these patients.

Standard laboratory methods were employed for complete blood count, using DxH 900 Hematology Analyzer produced by Beckman Coulter Inc., Brea, CA, USA. Biochemical parameters were measured with an AU5800 Chemistry Analyzer manufactured by Beckman Coulter Inc., Brea, CA, USA.

Serum levels of cholestasis-related parameters (AlkPh, GGT, and ConjBil, referred to as biliary parameters), along with AST and ALT, were employed to characterize liver–biliary function. Henceforth, they will be collectively designated as liver–biliary parameters (LBPs). As the Sequential Organ Failure Assessment (SOFA) scoring system takes into account total bilirubin (TotBil) instead of ConjBil, TotBil was also included in the univariate analysis. Neutrophil count, leukocyte count, and blood levels of presepsin, C reactive protein (CRP), and fibrinogen were used to assess the septic/inflammatory state. For the sake of completion, other parameters were also considered, including age, the variables employed for calculating SOFA score [thrombocyte count, Glasgow coma scale at admission (GCS_at_admission) at admission, hemodynamic status, and serum levels of TotBil and creatinine (and urea)], as well as parameters characterizing the hemodynamic and metabolic impact of the septic/inflammatory state (arterial blood pH and serum levels of bicarbonate, lactate, and NTproBNP). The categorical parameters taken into account were gender, outcome in terms of survival (deceased or not), and the type of infection (digestive, urinary, pulmonary, cutaneous).

A diagnosis of pulmonary infection was made in patients with progressive infiltrate, consolidation, or cavitation on chest imaging (radiography or computed tomography) associated with clinically (fever, sweating, cough with purulent sputum, dyspnea, rales on auscultation) and biologically (leukocytosis, increased inflammatory markers, elevated procalcitonin or presepsin) relevant manifestations [43]. In contrast to the urinary tract infections (UTI), we generally did not rely on microbiologic tests in the diagnosis of respiratory infection, as these have variable sensitivity and specificity and do not significantly influence the therapeutic approach to the patient, at least in the initial phases. Moreover, administrative constraints do not allow for obtaining sputum samples in our emergency department (although obtaining urine samples for culture is possible and commonly performed). Nonetheless, the antibiotic is started frequently in the emergency department a few hours after admission, as soon as a pulmonary infection is suspected (based on clinical, imaging, and laboratory data) as the cause of the septic condition. Sputum samples are obtained the next day, when the patient is already under antibiotic treatment—as a result, sputum cultures are usually negative. Sputum cultures are, however, routinely obtained from patients admitted to the intensive care unit, and they are primarily used to guide antibiotic treatment. Other clinicians appear to have a similar approach to the diagnosis and management of respiratory infections [44].

UTIs were diagnosed in patients with a urine culture (obtained from clean-catch urine samples) positive for a single organism, with bacterial growth greater than 100,000 CFUs/mL, accompanied by increased urine leukocyte count and/or positive urine leukocyte esterase test, along with biological markers of inflammation/infection (leukocytosis, increased inflammatory markers, elevated procalcitonin or presepsin). In none of our patients were the urine samples obtained by invasive methods; therefore, a bacterial growth of less than 100,000 CFUs/mL was not considered indicative of a UTI. The growth of multiple organisms was attributed to contamination and consequently did not qualify as evidence of a UTI. The presence of symptoms, suggestive of either upper (fever, chills, lumbar pain, nausea, vomiting) or lower (dysuria, urinary frequency and/or urgency, suprapubic discomfort) urinary tract involvement was not considered essential due to their frequent absence in elderly and critically ill patients [45].

Patients presenting with an abrupt onset of diarrhea (liquid stools persisting for >12 h), alongside nausea, vomiting, abdominal pain, or fever, in the absence of noninfectious conditions that could explain these symptoms, accompanied by an elevated blood leukocyte count and increased serum inflammatory markers, were diagnosed with digestive infection [46].

A diagnosis of cutaneous infection was established in patients with localized skin redness, edema, pain, and warmth complemented by signs and symptoms of systemic involvement: fever, leukocytosis, increased serum inflammatory markers, high procalcitonin or presepsin, in the absence of other obvious sources of infection [47].

### 2.4. Statistical Analysis

In this paper, the terms “variable” and “parameter” will be considered equivalent and used interchangeably, as will “association” and “correlation”.

The statistical analysis included descriptive and inferential statistics.

Descriptive statistics consisted of the calculation of the various quartiles (minimum, first quartile, median, third quartile, maximum) for the characterization of the continuous (numerical) parameters (age and parameters used to assess liver–biliary function, septic/inflammatory state, hemodynamic and metabolic status, including those employed in the calculation of SOFA score). The Shapiro–Wilk test was used to assess the normality of data distribution.

Inferential statistics encompassed univariate analysis (UVA) and multivariate analysis (MVA).

UVA included: (1) the calculation of the Spearman correlation coefficient when exploring the correlation between two numerical values, such as ConjBil, AST, ALT, AlkPh, GGT, presepsin, age, neutrophil, and leukocyte count; (2) Mann–Whitney test with continuity correction to compare continuous (numerical) with categorical parameters representing various dichotomous categories of patients, such as categories of outcome, gender, and type of infection. When multiple comparisons were performed, the significance level (commonly set at 0.05) was lowered according to Bonferroni correction: the corrected significance level was 0.05 divided by the number of comparisons. All the statistical calculations were performed using the R language and environment for statistical computing and graphics (version 4.2.3); the employed packages were tidyverse, ggplot2, mosaic, reshape2, and aod.

Despite the broad range of presepsin values, spanning several orders of magnitude (from 60 to over 20,000 pg/mL), the serum level of presepsin itself, rather than its logarithm, was utilized in the statistical calculations. This choice contrasts with that of other researchers [41,42]. The decision was made because logarithm is a monotonic increasing function, and the statistical calculations were conducted using non-parametric tests, which are unaffected by the data distribution (whether normal or otherwise), or the order of magnitude of the analyzed data.

After having established, by UVA, the parameters significantly associated with each of the LBPs, MVA was performed to determine which of these parameters maintain independent associations with each of the LBP. Backwards stepwise regression was used. For each dependent variable, regression analysis was first performed on all *n* (supposedly) independent variables identified by UVA as statistically significantly associated with the dependent variable (Appendix A).

Among the variables proven to lack a statistically significant (*p* > 0.05) contribution, the one with the lowest contribution (which was also the variable with the smallest T value) was eliminated, and the regression was performed again on the remaining *n* − 1 variables, and the one with the smallest contribution was eliminated. The procedure was continued until all remaining variables were statistically significantly associated with the dependent variable.

## 3. Results

Out of the 396 patients, 312 had a SOFA score ≥ 2 (which means sepsis according to The Third International Consensus Definitions for Sepsis and Septic Shock [1,2]) (191 with normal ConjBil, 121 with increased ConjBil), the remaining 84 patients had a SOFA score < 2 (62 with normal ConjBil, 22 with increased ConjBil). For reasons explained in the penultimate paragraph of Section 4, no statistical test (such as Fisher’s exact test) was applied to these numbers.

### 3.1. Univariate Analysis

The Shapiro–Wilk test demonstrated that no numerical (either continuous or discrete) parameter had a normal (Gaussian) distribution. Consequently, the Mann–Whitney test was used to investigate the association of numerical with categorical parameters. Moreover, as one cannot assume linear relationships between the various numerical variables, the Spearman method was employed to perform correlation/simple regression analysis when investigating the association of numerical with numerical parameters. At each stage of the statistical calculations, the problem of multiple comparisons was dealt with by applying Bonferroni correction. For example, referring to the computations in Table 1, the number of performed comparisons was 20; according to Bonferroni correction, the level of statistical significance should be lowered to 0.05/20 = 0.0025.

#### 3.1.1. Comparisons between Patients with Increased and Normal Conjugated Bilirubin

Comparisons were made between the relevant demographic, biological, and clinical parameters of patients with normal ConjBil levels (*n* = 253) and those with increased ConjBil levels (*n* = 143), as displayed in Table 1 (numerical variables) and Table 2 (categorical variables).

Besides the expected (and rather trivial) association with transaminase level, increased ConjBil levels showed statistically significantly associations only with presepsin and the parameters reflecting a compromised hemodynamic status, including lactate level and the cardiovascular parameter in the SOFA score. Among the categorical parameters, only male gender demonstrated a statistically significant association.

#### 3.1.2. Associations between Categorical and Numerical Parameters

The Mann–Whitney test revealed several meaningful findings. Deceased patients exhibited significantly higher levels of AST and ConjBil, but not TotBil. The levels of ConjBil and TotBil were statistically significantly higher in males and in patients with cutaneous and urinary infections, although the differences were not clinically significant (see Table 3).

Among categorical parameters, ConjBil showed stronger statistical associations compared to TotBil, except for urinary infection. The only comparison surviving Bonferroni correction for this more stringent threshold was between ConjBil and cutaneous infection. At this stage, the most probable association of ConjBil was found to be with presepsin, as it demonstrated the lowest *p*-value.

#### 3.1.3. Correlations between Numerical Parameters

Simple regression analysis using the Spearman method revealed many associations between the LBP and other numerical parameters, as detailed in Table 4 and Figure 2, which represent a heatmap of these correlations.

Only correlations with a *p*-value of 0.05 or lower are listed in Table 4. Due to the large number of comparisons (96), the threshold for statistical significance was adjusted using the Bonferroni correction to 0.05/96 ≈ 0.0005. The results meeting this more stringent threshold are highlighted in bold in Table 4. A highly statistically significant, low to medium strength, direct correlation was observed between presepsin and AlkPh, GGT, ALT, AST, ConjBil, and TotBil. Most other associations are related to hemodynamic status, including lactate, urea, creatinine, and the cardiovascular parameter in the SOFA score. As of this point, the primary focus continues to center on the correlations between LBP and presepsin, given their consistent display of the lowest *p*-values.

### 3.2. Multivariate Analysis

To determine which among these parameters independently influences the LBP levels, a multivariate analysis was performed. The analysis demonstrated that presepsin was indeed the factor with the highest probability of being associated with each of the cholestasis-related parameters, as shown in Table 5. This underscores the pivotal role of presepsin in influencing these associations and suggests that it offers valuable insights into real-world relationships.

## 4. Discussion

To our knowledge, this is the first attempt to employ MVA to identify factors independently associated with cholestasis-related parameters within a sepsis context. This exploration may offer insights into the pathology of biliary tract disorders triggered by systemic infection, potentially helping to guide clinical decision-making and avoid unnecessary investigations for alternative causes of cholestasis.

The practitioner often deals with patients in whom sepsis diagnosis is uncertain, and has to decide whether to prescribe antibiotics or not. Failure to timely recognize sepsis and to treat it promptly may seriously impact prognosis and increase mortality [48]. On the other hand, the abusive use of antibiotics leads to the selection of resistant bacterial strains [49]. Distinguishing septic conditions from non-septic ones can be challenging due to shared signs and symptoms. For example, hypovolemic shock might be confused with septic shock, as both can cause an increase in leukocytes [19,50] and can lead to liver hypoperfusion with consequent hepatic and biliary injury [51,52]. Also, in elderly patients presenting with acute decompensated heart failure, a high white blood cell count induced by acute stress [53] may be mistakenly considered a sign of infection, especially when X-ray findings are asymmetric due to the lateral decubitus positioning of the patient. Furthermore, the absence of fever, often seen in elderly individuals with infection, could be disregarded as evidence against infection [54]. Both acute decompensated heart failure and infections may result in cholestasis: infection through multiple mechanisms [32], and heart failure by inducing liver congestion [55,56]. While elevated inflammation markers may suggest infection, it is important to note that heart failure itself is associated with inflammation [57]. These markers also may not distinguish between bacterial and viral infections [58]. Consequently, markers with higher specificity for bacterial infection are needed, and presepsin stands out as one of them.

MVA revealed presepsin as the variable most probably associated with cholestasis-related parameters, being the only one correlated with all of them. In contrast, the parameters reflecting inflammatory (CRP), hemodynamic (serum urea and creatinine), and coagulation (thrombocyte count) status showed an association with fewer cholestasis-related parameters, and these associations were more likely to be spurious. This suggests that changes in the biliary system causing the intra-sepsis cholestasis are induced by the septic state per se, and not by the systemic inflammatory state, hemodynamic compromise, or coagulation disorders.

Other researchers have also found correlations between presepsin level and biliary parameters, further reinforcing these associations. A study performed on 567 patients showed that presepsin levels were correlated with biliary enzymes (both AlkPh and GGT) even in the absence of sepsis (and kidney dysfunction), MVA identifying SOFA score, AlkPh, and serum creatinine, identified as independently associated with presepsin level. As high levels of presepsin were identified not only in the blood but also in bile and Kupffer cells, it was hypothesized that excessive pressure in the biliary tract leads to increased presepsin expression in Kupffer cells, with spillover into the blood and bile [42]. Furthermore, a retrospective study on 1840 patients reported that presepsin is positively correlated with parameters indicative of cholestasis (including ConjBil, AlkPh, GGT) and AST, with the authors suggesting that this might be linked to the dual elimination of presepsin not only via the kidney but also through the hepatobiliary route [41]. A result somewhat similar to the present study was reached by an older one demonstrating by MVA that an AlkPh > 100 IU is among the few independent predictors of true bacteremia in adult hospitalized patients, alongside common factors such as temperature ≥ 39 °C, immunosuppressive treatment, and hospitalization in an intensive care unit [59]. A more recent study, also employing MVA, identified bilirubin, along with CRP, respiratory rate, and procalcitonin as factors independently associated with bacteremia [60]. In another study, UVA revealed AlkPh, GGT, and cholinesterase (but not bilirubin, transaminases, and CRP) as significantly associated with bacteremia. Although AlkPh appeared to have the most statistically significant association, MVA pointed out cholinesterase as the only factor independently associated with bacteremia [61].

While these studies concentrated on bacteremia and employed MVA to establish the factors independently associated with it, our research shifted its focus to cholestasis-related parameters. We employed MVA to pinpoint presepsin as the biological parameter with the highest probability of being associated with them. Despite the fact that correlation does not necessarily imply causation, this result strongly suggests that the infectious process should be the primary consideration when investigating the etiology of cholestasis in septic patients. Alternative explanations are to be sought only when telltale signs are present, such as dilated biliary ducts on imaging tests.

Attempts have also been made to determine whether AlkPh is associated not only with systemic infection but also with inflammation. A study of this nature demonstrated a connection between AlkPh and CRP in women (while absent in men), advancing the hypothesis that AlkPh might be a marker of systemic inflammation [62]. It is established that CD14 is expressed on the cells directly involved in combating infection, such as monocytes, macrophages, and neutrophils [4]. However, evidence is accumulating that hepatocytes, Kupffer cells [4], and biliary epithelial cells [63] might also serve as sources of sCD14, as they express CD14 under the influence of LPS [4,64]. Consequently, the resulting sCD14 binds LPSs and activates CD14-negative cells, such as endothelial and epithelial cells [4]. It is supposed that hepatocytes and Kupffer cells are involved in endotoxin clearance by producing LPS-binding sCD14 [65]. Other researchers have attributed an acute phase protein status to CD14, noting its correlation with CRP and IL6, which promotes its synthesis by hepatocytes [66].

Therefore, the simultaneous occurrence of cholestasis and sepsis may arise not only from a synchronous relationship, as seen in cases like cholangitis where they share a common etiology, but also from a sequential relationship where sepsis causally precedes cholestasis. The sequential relationship is typically more common than the synchronous one. Hence, when noticing increased serum levels of biliary enzymes (sometimes accompanied by conjugated hyperbilirubinemia) in a septic patient, the first suspicion should be towards a sequential relationship.

In our study, SOFA score was not considered an independent variable in MVA as it is itself a dependent variable, calculated based on five other parameters. However, all five parameters were considered in the MVA if the UVA revealed them as significantly associated with at least one of the cholestasis-related parameters. The MVA highlighted only two of them as independent correlates of the biliary parameters: serum creatinine for AlkPh and ConjBil, and thrombocyte count for ConjBil.

The reason we have not attempted to investigate putative correlations between cholestasis-related parameters, primarily ConjBil, and the SOFA score and/or sepsis is that ConjBil is a component of TotBil, which is in turn a component of the SOFA score, which itself is a component of the definition of sepsis [1,2]. Therefore, a positive correlation is expected between ConjBil and TotBil, between TotBil and the SOFA score, and between the SOFA score and sepsis. Since correlation is a transitive relation, a positive correlation is consequently expected between ConjBil and sepsis. This relationship is essentially a statistical truism, and seeking such correlations would be circular. Moreover, as ConjBil is strongly correlated with AlkPhos and GGT, correlations between these enzymes and sepsis would also be trivial. Hence, this study did not endeavor to explore such associations.

The two most important limitations of the present study are its retrospective nature and the inclusion of the patients based on having a presepsin measurement performed at admission. Another limitation lies in the relatively low number of patients; however, the statistically strong associations revealed by our study made this limitation of little consequence. An important drawback is the absence of interleukins and oxidative stress markers among the biological parameters considered and analyzed. Certainly, only a prospective study would be capable of overcoming this limitation.

Looking ahead, a prospective study, preferably multicentric, should be conducted with careful consideration given to the challenges associated with measuring presepsin. It would be impractical to determine presepsin levels in all patients presenting to the emergency department, as this could lead to an overcrowded sample of patients with mostly irrelevant data. One viable alternative would be to measure presepsin in patients suspected of sepsis. Therefore, clear criteria for raising suspicion of sepsis should be established and consistently applied. A broader set of biological parameters, such as cytokines and markers of oxidative stress, could be encompassed.

## 5. Conclusions

Sepsis sets in motion a myriad of events, including the activation of biological pathways, the up- or downregulation of various enzymes and transport systems, and the impairment of organ function, particularly in the liver and kidneys. This leads to increased levels of both cholestasis-related parameters (such as ConjBil and biliary enzymes) and presepsin (Figure 3). Consequently, as indicated by our research, cholestasis-related parameters are associated with presepsin with a higher probability than with hemodynamic, inflammatory, or coagulation-related variables in a septic state. This finding holds promise for limiting imaging investigations in emergency patients, enabling clinicians to prioritize treatment of the primary infection.

## Figures and Tables

**Figure 1 diagnostics-14-01706-f001:**
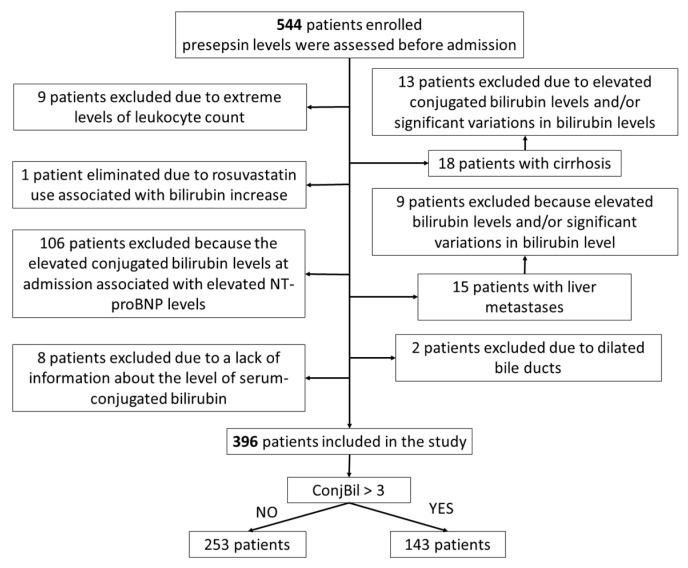
Flowchart detailing the patient selection process; NTproBNP—N-terminal pro B-type natriuretic peptide; ConjBil—conjugated bilirubin.

**Figure 2 diagnostics-14-01706-f002:**
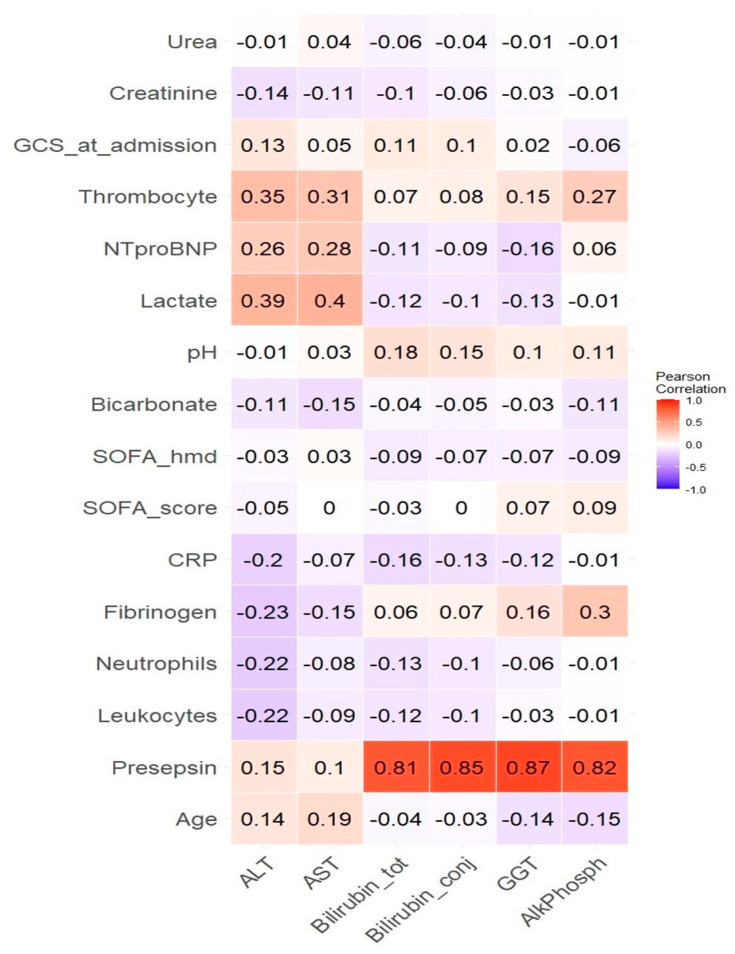
Heatmap of the correlations between the liver–biliary parameters (considered dependent variables) and the other numerical parameters (considered independent variables) as calculated by simple regression. The numerical values are the correlation coefficients. AlkPhosph: alkaline phosphatase; ALT: alanine aminotransferase; AST: aspartate aminotransferase; Bilirubin_conj: conjugated bilirubin; Bilirubin_tot: total bilirubin; CRP: C reactive protein; GGT: Gamma-glutamyl transferase; GCS_at_admission: Glasgow coma scale at admission; NTproBNP: N-terminal prohormone of brain natriuretic peptide; SOFA_score: Sequential Organ Failure Assessment score; SOFA_hmd = the fourth parameter in the SOFA score reflecting the hemodynamic status (Mean arterial pressure OR administration of vasoactive agents required).

**Figure 3 diagnostics-14-01706-f003:**
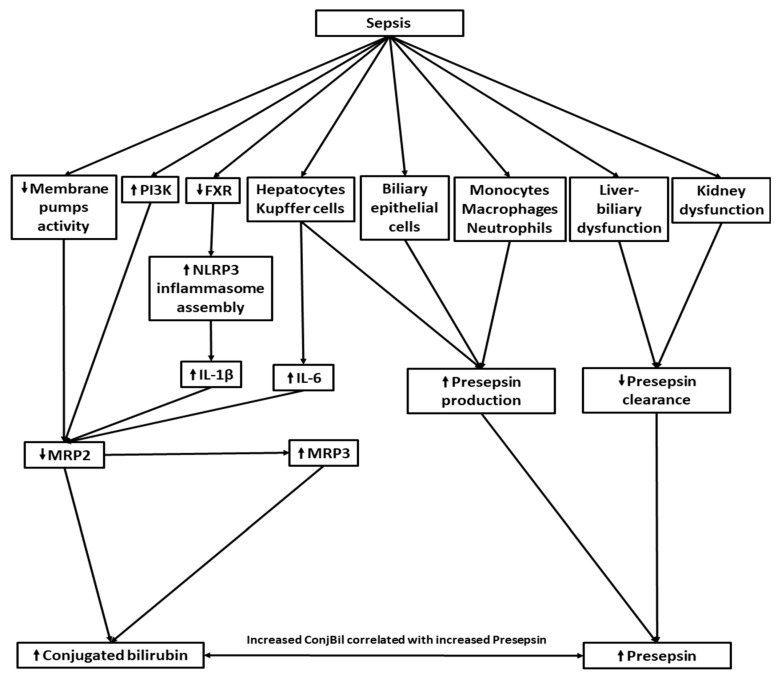
Correlation between conjugated bilirubin and presepsin in sepsis. Sepsis induces cholestasis by reducing the activity of membrane pumps, including MRP2, which is involved in transporting bilirubin from hepatocytes into the bile ducts. In sepsis, the PI3K signaling pathway is activated, leading to MRP2 reduction. The downregulation of the Farnesoid X receptor further contributes to NLRP3 inflammasome assembly, IL-1β synthesis, and subsequent MRP2 decrease. Increased MRP3 activity serves as a protective mechanism, facilitating the transport of excess bilirubin from hepatocytes into the bloodstream. LPS stimulates hepatocytes and Kupffer cells to produce IL-6, which reduces MRP2 levels. In sepsis, various cells (including monocytes, macrophages, and neutrophils, along with Kupffer cells, hepatocytes, and biliary epithelial cells) stimulated by LPS increase presepsin production. However, kidney and liver biliary dysfunction induced by sepsis decreases presepsin clearance, resulting in elevated presepsin levels. ↓: decrease/downregulate, ↑: increase/upregulate ConjBili: conjugated bilirubin, IL: interleukin, FXR: Farnesoid X receptor, LPS: lipopolysaccharides, MRP2: multidrug resistance protein 2, PI3K: phosphatidylinositol-3-kinase.

**Table 1 diagnostics-14-01706-t001:** Comparison between the numerical variables, which reflect demographic and biological features, characterizing two groups of patients: those with normal conjugated bilirubin (*n* = 253) vs. those with increased conjugated bilirubin (*n* = 143).

Numerical Parameter	Median (IQR) for Patients with Increased ConjBil	Median (IQR) for Patients with Normal ConjBil	W Statistics	*p*-Value
Age (years)	73 (65–82.5)	73 (64–82)	17,555	0.6
**Presepsin (pg/mL)**	**1340 (618.5–3220.5)**	**789 (318–1700)**	**12,633.5**	**6 × 10^−7^**
Leukocytes (μL)	16,120 (9995–21,740)	14,660 (10,620–21,200)	17,389.5	0.5
Neutrophils (μL)	12,665 (7695–18,654.5)	12,009.5 (8057.5–18,132.25)	17,299.5	0.5
Fibrinogen (mg/dL)	495.5 (383–757)	482 (385–660)	15,899.5	0.2
C reactive protein (mg/dL)	16.05 (8.25–27.75)	9.4 (3.7–20.8)	3612.5	0.006
**SOFA_score**	**5 (2–8)**	**4 (2–6)**	**14,210**	**4 × 10^−4^**
**SOFA_hmd**	**0 (0–2)**	**0 (0–0)**	**15,112**	**4 × 10^−4^**
Bicarbonate (mEq/L)	19 (15.9–21.4)	20.5 (15.175–24.5)	17,459.5	0.02
pH	7.42 (7.3–7.48)	7.37 (7.28–7.45)	12,653	0.008
**Lactate (mmol/L)**	**2.18 (1.215–4.005)**	**1.27 (0.8–2.23)**	**6794.5**	**5 × 10^−5^**
NTproBNP (pg/mL)	226 (223–325)	4710 (1641.5–15,948)	450.5	0.05
Thrombocyte count (× 10^3^/μL)	225 (167–310)	261 (195–351)	21,040	0.007
GCS_at_admission	15 (10–15)	15 (11–15)	19,428	0.2
Creatinine (mg/dL)	2.1 (1.065–3.725)	1.62 (0.94–4.18)	17,063	0.3
Urea (mg/dL)	92 (54–155)	69 (42–155)	16,050	0.06
**ALT (U/L)**	**41 (25–78.5)**	**28 (17–45)**	**12,695**	**8 × 10^−7^**
**AST (U/L)**	**49 (28–102)**	**28 (20–43)**	**11,134**	**2 × 10^−10^**
GGT (U/L)	51 (32.5–126)	41 (24.55–76.5)	6143.5	0.02
AlkPh (U/L)	101.5 (76–147.25)	83 (66–119)	5397	0.004

The Mann–Whitney test was used to calculate W statistics and *p*-values. According to Bonferroni’s correction, the threshold for the *p*-value should be set at 0.05/20 (as 20 comparisons were conducted), resulting in a threshold of 0.0025. Statistically significant results are indicated by bold typing. AlkPh: alkaline phosphatase; ALT: alanine aminotransferase; AST: aspartate aminotransferase; ConjBil: conjugated bilirubin; GGT: gamma-glutamyl transferase; GCS_at_admission: Glasgow coma scale at admission; IQR: interquartile range; NTproBNP: N-terminal prohormone of brain natriuretic peptide; SOFA_score: Sequential Organ Failure Assessment score; SOFA_hmd = the fourth parameter in the SOFA score reflecting the hemodynamic status (Mean arterial pressure OR administration of vasoactive agents required).

**Table 2 diagnostics-14-01706-t002:** Comparison between the categorical parameters, which reflect demographic and clinical features, characterizing two groups of patients: those with normal conjugated bilirubin (*n* = 253) vs. those with increased conjugated bilirubin (*n* = 143).

Categorical Parameter	a/b/c/d	Odds Ratio (95% CI)	*p*-Value
Pulmonary infection	197/56/110/33	1.06 (0.62 to 1.77)	0.9
Urinary infection	201/52/105/38	1.4 (0.84 to 2.32)	0.17
Cutaneous infection	248/5/133/10	3.72 (1.13 to 14.16)	0.024
Digestive infection	247/6/141/2	0.58 (0.06 to 3.33)	0.72
Gallbladder stones	230/23/130/13	1 (0.45 to 2.14)	1
**Male gender**	**154/99/60/83**	**2.15 (1.39 to 3.34)**	**0.00035**
Deceased	163/90/82/61	1.35 (0.87 to 2.09)	0.2

Fisher’s exact test was used to calculate *p*-values. According to Bonferroni’s correction, the threshold for the *p*-value should be set at 0.05/7 (as 7 comparisons were performed), namely 0.007. Consequently, the *p*-value is considered significant only for the male gender (highlighted in bold), indicating that men are more prone to ConjBil elevation. a = number of patients with normal ConjBil and a negative value of the categorical parameter (e.g., pulmonary infection absent), b = number of patients with normal ConjBil and a positive value of the categorical parameter (e.g., pulmonary infection present), c = number of patients with increased ConjBil and a negative value of the categorical parameter, d = number of patients with increased ConjBil and a positive value of the categorical parameter. For example, in the second row and second column, 197 represents the number of patients with normal ConjBil who were free of pulmonary infection, while 56 denotes those with normal ConjBil who had pulmonary infection. 95% CI = 95% confidence interval of odds ratio.

**Table 3 diagnostics-14-01706-t003:** Results of the Mann–Whitney test regarding the association between the numerical parameters of liver–biliary function and the categorical parameters. Only the results with a *p*-value < 0.05 are displayed.

Numerical Parameter	Categorical Parameter	#Pts	Median (IQR)	#Pts	Median (IQR)	W Statistics	*p*-Value
		Deceased		Survivors			
AlkPh	Survival	151	106 (78–131)	245	85 (66–120.25)	4751	0.01
AST	Survival	151	37 (24–68.5)	245	30 (20–52)	15,880.5	0.02
ConjBil	Survival	151	0.26 (0.2–0.515)	245	0.24 (0.16–0.39)	15,377.5	0.005
		Male gender		Female gender			
ConjBil	Gender	182	0.28 (0.19–0.5)	214	0.22 (0.16–0.36)	15,948.5	0.002
TotBil	Gender	182	0.66 (0.49–1.1275)	214	0.575 (0.44–0.96)	17,239.5	0.05
		Cutaneous infection_yes		Cutaneous infection_no			
**ConjBil**	**Cutaneous infection**	**15**	**0.59 (0.265–1.775)**	**381**	**0.25 (0.17–0.41)**	**1383**	**0.0007**
TotBil	Cutaneous infection	15	0.91 (0.605–2.445)	381	0.62 (0.45–1.02)	1546	0.003
		Urinary infection_yes		Urinary infection_no			
ConjBil	Urinary infection	90	0.28 (0.2–0.465)	306	0.24 (0.16–0.41)	11,740	0.03
TotBil	Urinary infection	90	0.73 (0.523–1.078)	306	0.605 (0.44–1.015)	11,508.5	0.02

According to Bonferroni’s correction, the threshold for the *p*-value should be set at 0.05/42 as 42 comparisons were performed (6 numerical parameters × 7 categorical parameters); therefore, the threshold is set at 0.001. Consequently, only comparison between ConjBil and cutaneous infection was statistically significantly (highlighted in bold) #Pts = patients count; AlkPh: alkaline phosphatase; AST: aspartate aminotransferase; ConjBil: conjugated bilirubin; IQR: interquartile range; TotBil: total bilirubin.

**Table 4 diagnostics-14-01706-t004:** Cross-correlation table between liver–biliary parameters (considered dependent variables) and the other numerical parameters (considered independent variables) using simple regression analysis.

	ALT	AST	TotBil	ConjBil	GGT	AlkPh
Age					**0.00014**	0.049
Bicarbonate		0.00082		0.0031		0.0042
Creatinine		**0.00035**		0.048		0.0033
CRP		0.05	0.028	0.0015		0.027
Fibrinogen						0.04
GCS_at_admission		**0.00043**				
Lactate	0.0085	**4.3 × 10^−7^**	**0.00041**	**0.00017**		0.0084
Leukocytes						0.012
Neutrophils						0.003
NTproBNP						0.013
pH			**0.00018**	0.0059		
Presepsin	**7.7 × 10^−6^**	**3.9 × 10^−12^**	**0.00014**	**3 × 10^−11^**	**2.7 × 10^−8^**	**1.7 × 10^−11^**
SOFA_hmd		0.0064	0.0055	**9.3 × 10^−6^**		0.0027
SOFA_score	0.0025	**2.3 × 10^−8^**	0.0022	**2.2 × 10^−5^**		0.0036
Thrombocyte count			**9.6 × 10^−6^**	0.011		
Urea	0.042	**3.5 × 10^−5^**		**0.00035**		0.0025

To increase readability, only the *p*-values ≤ 0.05 are shown. Given the probable non-linear relationship between these numerical parameters, the Spearman method was employed for performing the calculations. Due to the large number of comparisons (96), the threshold for statistical significance was lowered to 0.05/96 ≈ 0.0005. The results meeting this threshold are indicated in bold type. AlkPh: alkaline phosphatase; ALT: alanine aminotransferase; AST: aspartate aminotransferase; ConjBil: conjugated bilirubin; CRP: C reactive protein; GGT: Gamma-glutamyl transferase; GCS_at_admission: Glasgow coma scale at admission; IL: interleukin; NTproBNP: N-terminal prohormone of brain natriuretic peptide; SOFA_score: Sequential Organ Failure Assessment score; SOFA hmd: the fourth parameter in the SOFA score [“Mean arterial pressure OR administration of vasoactive agents required”; may have discrete (integer) values 1 to 4]; TotBil: total bilirubin.

**Table 5 diagnostics-14-01706-t005:** Multivariate analysis. In the first column are the cholestasis-related parameters, AST, and ALT, while the second column lists the factors independently associated with them.

Dependent Variable	Independent Variable	Estimate	95% Confidence Interval	t-Statistic	*p*-Value
AlkPh	Presepsin	0.019	(0.012, 0.024)	5.649	7 × 10^−8^
AlkPh	Creatinine	−7.58	(−12.98, −2.18)	−2.752	0.007
AlkPh	CRP	1.34	(0.2, 2.48)	2.300	0.02
ALT	Lactate	8.97	(4.65, 13.28)	4.074	6 × 10^−5^
ALT	Presepsin	0.0095	(0.0048, 0.014)	3.957	1 × 10^−4^
AST	Lactate	13.6	(6.26, 20.93)	3.631	0.0003
AST	Presepsin	0.013	(0.0055, 0.021)	3.331	0.001
AST	SOFA_hmd	26.95	(6.28, 47.61)	2.555	0.01
ConjBil	Presepsin	0.00011	(0.000080, 0.00013)	8.185	4 × 10^−15^
ConjBil	Creatinine	−0.077	(−0.11, −0.044)	−4.665	4 × 10^−6^
ConjBil	Urea	0.0014	(0.00040, 0.0024)	2.767	0.006
ConjBil	Thrombocyte count	0.00043	(−0.00082, −0.000032)	−2.117	0.035
GGT	Presepsin	0.019	(0.013, 0.024)	6.480	5 × 10^−10^
GGT	Age	−1.76	(−2.90, −0.61)	−3.004	0.003

The third column contains the estimate, which is the average change in the log odds of the dependent variable associated with a one-unit increase in each independent variable. For each parameter in the first two columns (except SOFA_hmd, thrombocyte count, and age) the name of the parameter should be understood as the serum level of the respective parameter. AlkPh: alkaline phosphatase; ALT: alanine aminotransferase; AST: aspartate aminotransferase; ConjBil: conjugated bilirubin; CRP: C reactive protein; GGT: gamma-glutamyl transferase; SOFA hmd: the fourth parameter in the SOFA score [“Mean arterial pressure OR administration of vasoactive agents required”; may have discrete (integer) values 1 to 4].

## Data Availability

All data used for performing the computations and the generation of graphs are available upon request to the corresponding author (D.D., dorin.dragos@umfcd.ro).

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
