# Peer review of "The Pivotal Role of Presepsin in Assessing Sepsis-Induced Cholestasis"

_diagnostics, 2024, doi:10.3390/diagnostics14161706_

Round 1

Reviewer 1 Report (Previous Reviewer 2)

Comments and Suggestions for Authors

The manuscript is well written with clear figures and tables, the statistical analysis was approached excellently, satisfying all the results obtained.

Author Response

Reviewer 1

The manuscript is well written with clear figures and tables, the statistical analysis was approached excellently, satisfying all the results obtained.

Response: Thank you very much for your time and positive feedback on our submission.

Reviewer 2 Report (New Reviewer)

Comments and Suggestions for Authors

Thank you for allowing me to read and review this excellent paper! It is scientifically sound, and I believe it will have good clinical impact if published.

Line 64: lateral decubitus is a position. Thus, please state “…. due to the lateral decubitus positioning of the patient.”

Why is the text from lines 73-84 in red colour?

The introduction is too lengthy. Probably some text here can be shifted to the discussion section.

Line 107: is there not supposed to be a comma between “increases” and “both” in the statement “As sepsis increases, both presepsin and cholestasis-related parameters, a positive correlation between presepsin level and biliary parameters is to be expected?”

Under section “Study design and ethical issues,” the authors mentioned that informed consent was obtained (line 128) when the study design was retrospective rather than prospective (line 125). Please confirm if this was indeed the case as some patients were even deceased (line 308).

Fig 1 does not have a footer explanation on the abbreviations used (e.g. NT-proBNP and conjbil)

Line 184: were cultures of respiratory specimens not looked at when diagnosing pulmonary infections?

Line 189: it is well-known that a positive urine culture does not necessary indicate a UTI. For instance, a sample which grew two or more different organisms is considered contaminated. Thus, please refine your UTI diagnosis criteria.

The discussion section was very well-written. In particular, the authors attempted to scientifically explain why presepsin was the variable most likely to be associated with cholestasis-related parameters and they have managed to defend their study decisions (such as not investigating correlations between cholestasis-related parameters and the SOFA score).

Author Response

Reviewer 2

Comments 1: Thank you for allowing me to read and review this excellent paper! It is scientifically sound, and I believe it will have good clinical impact if published.

Response 1: Thank you very much for your constructive feedback and positive assessment of our study. Please find below a detailed response to your comments. The changes made in the manuscript are highlighted in red.

Comments 2: Line 64: lateral decubitus is a position. Thus, please state “…. due to the lateral decubitus positioning of the patient.”

Response 2: The change was made as indicated.

Comments 3: Why is the text from lines 73-84 in red colour?

Response 3: This is a carryover from a previous revision; we have corrected it in the manuscript.

Comments 4: The introduction is too lengthy. Probably some text here can be shifted to the discussion section.

Response 4: A significant paragraph from the Introduction was moved to the Discussion section (highlighted in yellow), as suggested.

Comments 5: Line 107: is there not supposed to be a comma between “increases” and “both” in the statement “As sepsis increases, both presepsin and cholestasis-related parameters, a positive correlation between presepsin level and biliary parameters is to be expected?”

Response 5: Thank you for bringing this confusing phrasing to our attention. We have revised it to: “As both presepsin and cholestasis-related parameters increase during sepsis, a positive correlation between presepsin level and biliary parameters (ConjBil, GGT, and AlkPh) in septic patients is to be expected”

Comments 6: Under section “Study design and ethical issues,” the authors mentioned that informed consent was obtained (line 128) when the study design was retrospective rather than prospective (line 125). Please confirm if this was indeed the case as some patients were even deceased (line 308).

Response 6: We have included the following text in the “2.1. Study design and ethical issues” subsection: All patients admitted to our hospital are asked to sign an informed consent form, allowing the results from their clinical examination, laboratory tests, and imaging investigations to be used for research purposes. If patients are unable to sign due to cognitive impairment, impaired consciousness, or motor function limitations, a close relative is asked to sign on their behalf.

Comments 7: Fig 1 does not have a footer explanation on the abbreviations used (e.g. NT-proBNP and conjbil)

Response 7: The abbreviations used are now defined in the figure legend, as requested.

Comments 8: Line 184: were cultures of respiratory specimens not looked at when diagnosing pulmonary infections?

Response 8: As an answer to this pertinent question, we have inserted the following paragraph in the Methods section:

“By contrast to the UTI, we generally did not rely on microbiologic tests in the diagnosis of respiratory infection as these have variable sensitivity and specificity and do not influence significantly the therapeutic approach to the patient, at least in the initial phases. Moreover, administrative constraints do not allow obtaining sputum samples in our emergency department (although obtaining urine samples for culture is possible and commonly performed). Nonetheless, the antibiotic is started as soon as a pulmonary infection is suspected (based on clinical, imaging, and laboratory data) as the cause of the septic condition, frequently in the emergency department a few hours after admission. Sputum samples are obtained only the next day when the patient is already under antibiotic treatment – as a result, sputum cultures are usually negative. Sputum cultures are however routinely obtained from patients admitted to the intensive care unit and they are primarily used to guide antibiotic treatment. Other clinicians appear to have a similar approach to the diagnosis and management of respiratory infections.”

Comments 9: Line 189: it is well-known that a positive urine culture does not necessary indicate a UTI. For instance, a sample which grew two or more different organisms is considered contaminated. Thus, please refine your UTI diagnosis criteria.

Response 9: We fully agree with this pertinent observation and have made the following changes to the text: “UTI were diagnosed in patients with a urine culture (obtained from clean-catch urine samples) positive for a single organism, with bacterial growth greater than 100,000 CFUs/mL, accompanied by increased urine leukocyte count and/or positive urine leukocyte esterase test, along with biological markers of inflammation/infection (leukocytosis, increased inflammatory markers, elevated procalcitonin or presepsin). In none of our patients were the urine samples obtained by invasive methods, therefore a bacterial growth of less than 100,000 CFUs/mL was not considered indicative of UTI. The growth of multiple organisms was attributed to contamination and consequently did not qualify as evidence of UTI.”

Comments 10: The discussion section was very well-written. In particular, the authors attempted to scientifically explain why presepsin was the variable most likely to be associated with cholestasis-related parameters and they have managed to defend their study decisions (such as not investigating correlations between cholestasis-related parameters and the SOFA score).

Response 10: Thank you for your appreciation!

This manuscript is a resubmission of an earlier submission. The following is a list of the peer review reports and author responses from that submission.

Round 1

Reviewer 1 Report

Comments and Suggestions for Authors This retrospective clinical study was conducted on 544 patients and it isn’t well-designed. The limitation of this study is the small number of patients and a single-center retrospective research. Additionally, most of the sources cited as references are old (20-25 years old). Presepsin is not a current marker. In addition, I have stated some of my specific comments below.
1) The role of Presepsin in sepsis-induced cholestasis is emphasized in this manuscript. 2)The article does not address a specific gap in the field. 3) The content of the article does not contribute to the literature compared to other previously written articles. 4) It is not enough to interpret Sepsis with Presepsin alone. Therefore, a study with another biomarker with proven effectiveness (e.g. Procalcitonin) should be performed together with presepsin. 5) References are not appropriate. The number of current references is quite small. 6) The number of paintings is large and their quality is low. The number of tables needs to be reduced. 7) Again, although the 48th and 49th sources are shown as references in the discussion section, the number of sources in the reference section is 45. There is an inconsistency. 8) This study should be planned prospectively and other current markers should be used in addition to Presepsin. Therefore, more comprehensive and multicenter studies are needed for generalization.
In conclusion, this study does not add any new information to the current literature. This manuscript is not acceptable in its current form.

Author Response

Reviewer 1

This retrospective clinical study was conducted on 544 patients and it isn’t well-designed. The limitation of this study is the small number of patients and a single-center retrospective research.

Response: These two limitations of our study were addressed in the Discussion section: "The two most important limitations of the present study are its retrospective nature and [...]. Another limitation lies in the relatively low number of patients". However, the latter limitation (the relatively low number of patients) was mitigated by the strength of the statistical associations revealed by our study, as emphasized in the Discussion section. The rationale behind our study design is further elaborated in the following responses to the reviewer’s comments.

Additionally, most of the sources cited as references are old (20-25 years old).

Response: We have corrected this shortcoming; in the revised form of the article, the majority of references are at most 5 years old. The reasoning behind choosing old references was to address seminal sources.

Presepsin is not a current marker.

Response: We are uncertain about the reviewer's meaning of "not a current marker" and apologize for any confusion; it is unclear whether the reviewer considers it an outdated sepsis marker or one that is not yet established or validated.

To address the reviewer's concern, we have added the following paragraph to the Introduction section, with relevant references:

"Presepsin is a biomarker useful for the early diagnosis of sepsis [22–26], with a predictive ability for septic shock and organ dysfunction severity independent of procalcitonin [27] and an efficiency comparable to that of procalcitonin [22,28–30]. Presepsin has the advantage of increasing earlier than procalcitonin, allowing rapid detection, and is therefore particularly suited for use in the emergency department and critical care setting [31]. Some studies suggest that, compared to procalcitonin, presepsin has a higher sensitivity [32], a higher positive predictive value [11], and a similar area under the curve [33] for sepsis diagnosis, with a better predictive ability, particularly after hepato-biliary-pancreatic surgery [34]. Presepsin is also able to predict sepsis-related mortality in various settings [35,36], including intensive care units [37] and acute kidney injury [38]. It is a better predictor of sepsis related mortality than procalcitonin [39], with similar sensitivity but better specificity [40]."

In addition, I have stated some of my specific comments below.

1) The role of Presepsin in sepsis-induced cholestasis is emphasized in this manuscript.

Response: Indeed, our study shows a statistically significant association between presepsin levels and cholestasis-related parameters in septic patients, with presepsin being the only marker correlated with all of the investigated parameters in this category.  The increase in cholestasis-related parameters is associated with presepsin with a higher probability than hemodynamic, inflammatory, or coagulation-related variables. This robust link between sepsis and cholestasis, indicating cholestasis as being the consequence of sepsis, could eliminate unnecessary imaging procedures in critically ill patients, enabling clinicians to focus efforts on addressing the primary infectious cause.

2)The article does not address a specific gap in the field. 3) The content of the article does not contribute to the literature compared to other previously written articles.

Response: As stated in the first paragraph of the Discussion section: "To our knowledge, this is the first attempt to employ MVA in identifying factors independently associated with cholestasis-related parameters within a sepsis context." As far as we could ascertain from the scientific literature on this topic, other researchers have addressed different aspects of the cholestasis-sepsis relationship. We decided to focus on the determinants of the parameters that define cholestasis and have demonstrated that presepsin is the factor most closely associated with these parameters. This result, as far as we are aware, has not been published in any other study.

4) It is not enough to interpret Sepsis with Presepsin alone. Therefore, a study with another biomarker with proven effectiveness (e.g. Procalcitonin) should be performed together with presepsin.

Response: We thank the reviewer for this pertinent comment; indeed, including more biomarker candidates would have strengthened our results, and it is a valuable suggestion for future prospective studies, as we mentioned in the final part of the Discussions. However, we believe our research design is appropriate, especially considering that, according to available sources, presepsin is at least as effective in diagnosing sepsis as procalcitonin, as highlighted in the newly added (and cited above) paragraph of the Introduction.

(Presepsin is a biomarker useful for the early diagnosis of sepsis [22–26], with a predictive ability for septic shock and organ dysfunction severity independent of procalcitonin [27] and an efficiency comparable to that of procalcitonin [22,28–30]. Presepsin has the advantage of increasing earlier than procalcitonin, allowing rapid detection, and is therefore particularly suited for use in the emergency department and critical care setting [31]. Some studies suggest that, compared to procalcitonin, presepsin has a higher sensitivity [32], a higher positive predictive value [11], and a similar area under the curve [33] for sepsis diagnosis, with a better predictive ability, particularly after hepato-biliary-pancreatic surgery [34].)

5) References are not appropriate. The number of current references is quite small.

Response: As requested, we have increased the number of references to 66.

6) The number of paintings is large and their quality is low.

Response: The manuscript includes three figures:

- the first figure is a flowchart detailing the patient selection process, therefore we believe it cannot be eliminated;

- the second figure is a heatmap; while we could consider eliminating it, doing so would likely decrease the readability of the Results section;

- the third figure summarizes the conclusions of our study; if the reviewer deems it of low relevance, we are open to submitting it as supplementary material, although we believe it contributes to a better overview of the study’s message.

 > We have increased the resolution of the figures, aiming to improve their legibility as requested.

The number of tables needs to be reduced.

Response: We have eliminated the first table and submitted it as supplementary material.

The second table displays a general biological profile of the two study groups, which is typically regarded as  mandatory.

The third, fourth, and fifth tables display the results of the univariate analysis and therefore are essential for illustrating the core results. While we could condense them into a single table, doing so would reduce the readability of the results section. These univariate analysis results (displayed in the third, fourth, and fifth tables) are essential for informing the multivariate analysis, and identifying parameters to be included as putative independent determinants. The parameters were listed in Table 1, relegated to the status of Supplementary Table 1 in the revised manuscript.

The sixth table displays the results of the multivariate analysis and serves as the keystone of the Results section.

7) Again, although the 48th and 49th sources are shown as references in the discussion section, the number of sources in the reference section is 45. There is an inconsistency.

Response: This was a regrettable mistake, which we have now corrected. Thank you for bringing it to our attention.

8) This study should be planned prospectively and other current markers should be used in addition to Presepsin. Therefore, more comprehensive and multicenter studies are needed for generalization.

Response: We have addressed these aspects in the final paragraph of the Discussion section, where we also pointed out some of the difficulties incurred by undertaking a prospective multicentric study:

"Looking ahead, a prospective study, preferably multicentric, should be conducted with careful consideration given to the challenges associated with measuring presepsin. It would be impractical to determine presepsin levels in all patients presenting to the Emergency Department, as this could lead to an overcrowded sample of patients with mostly irrelevant data. One viable alternative would be to measure presepsin in patients suspected of sepsis. Therefore, clear criteria for raising suspicion of sepsis should be established and consistently applied. A broader set of biological parameters, such as cytokines and markers of oxidative stress, could be encompassed."

Based on our findings, efforts are justified for undertaking multicenter prospective studies in the future.

In conclusion, this study does not add any new information to the current literature.

Response: It is worth noting, however, that new insights have emerged: presepsin is the factor most closely associated with cholestasis-related parameters, demonstrating a stronger correlation with them than with hemodynamic, inflammatory, or coagulation-related variables in a septic state. To our knowledge, no other study has demonstrated the strength of this association or highlighted its potential for refining the diagnosis of critically ill patients in this category.

Reviewer 2 Report

Comments and Suggestions for Authors

The authors evaluate a possible pivotal role of Presepsin in Assessing Sepsis-Induced Cholestasis. In a sepsis setting, the increase in cholestasis related parameters is associated with presepsin with a higher probability than hemodynamic, inflammatory, or coagulation-related variables. These results can be useful for limiting imaging investigations in emergency patients and focusing the clinician on prioritizing primary infection treatment.

The manuscript is well written with clear figures and tables, the statistical analysis was approached excellently, satisfying all the results obtained.

The manuscript can be accepted in the current form.

Author Response

Reviewer 2

The authors evaluate a possible pivotal role of Presepsin in Assessing Sepsis-Induced Cholestasis. In a sepsis setting, the increase in cholestasis related parameters is associated with presepsin with a higher probability than hemodynamic, inflammatory, or coagulation-related variables. These results can be useful for limiting imaging investigations in emergency patients and focusing the clinician on prioritizing primary infection treatment.

The manuscript is well written with clear figures and tables, the statistical analysis was approached excellently, satisfying all the results obtained.

The manuscript can be accepted in the current form.

Response:  We appreciate the reviewer's time and effort in evaluating the manuscript, as well as the positive feedback on our study.